# Beneficial Use Impairments, Degradation of Aesthetics, and Human Health: A Review

**DOI:** 10.3390/ijerph19106090

**Published:** 2022-05-17

**Authors:** Erik D. Slawsky, Joel C. Hoffman, Kristen N. Cowan, Kristen M. Rappazzo

**Affiliations:** 1Oak Ridge Associated Universities (ORAU) Student Services Contractor at US EPA, Research Triangle Park, NC 27711, USA; slawsky.erik@epa.gov; 2Center for Computational Toxicology & Exposure, Office of Research and Development, US Environmental Protection Agency, Duluth, MN 55804, USA; hoffman.joel@epa.gov; 3Oak Ridge Institute for Science and Education (ORISE) United States Environmental Protection Agency (US EPA), Research Triangle Park, NC 27711, USA; cowan.kristen@epa.gov or; 4Department of Epidemiology, Gillings School of Global Public Health, University of North Carolina at Chapel Hill, Chapel Hill, NC 27514, USA; 5Center for Public Health & Environmental Assessment, Office of Research and Development, US Environmental Protection Agency, Research Triangle Park, NC 27711, USA

**Keywords:** beneficial use impairments, Great Lakes, green/bluespace, aesthetic degradation

## Abstract

In environmental programs and blue/green space development, improving aesthetics is a common goal. There is broad interest in understanding the relationship between ecologically sound environments that people find aesthetically pleasing and human health. However, to date, few studies have adequately assessed this relationship, and no summaries or reviews of this line of research exist. Therefore, we undertook a systematic literature review to determine the state of science and identify critical needs to advance the field. Keywords identified from both aesthetics and loss of habitat literature were searched in PubMed and Web of Science databases. After full text screening, 19 studies were included in the review. Most of these studies examined some measure of greenspace/bluespace, primarily proximity. Only one study investigated the impacts of making space quality changes on a health metric. The studies identified for this review continue to support links between green space and various metrics of health, with additional evidence for blue space benefits on health. No studies to date adequately address questions surrounding the beneficial use impairment degradation of aesthetics and how improving either environmental quality (remediation) or ecological health (restoration) efforts have impacted the health of those communities.

## 1. Introduction

The OneHealth approach recognizes the importance of interconnections between human health and the environment [1,2,3]. An exemplary international effort of this nature is the long-standing U.S.-Canada Great Lakes Water Quality Agreement (GLWQA), which aims to restore and protect roughly 21% of the global surface freshwater supply. Since its inception in 1972, the U.S.-Canada GLWQA has set forth the bi-national environmental and scientific goals to protect and restore the Laurentian Great Lakes (hereafter, Great Lakes). Through the Area of Concern (AOC) program under the auspices of the GLWQA, federal, provincial, state, tribal, and first nation partner agencies in the United States and Canada have laid the groundwork for identifying, remediating, and restoring degraded areas and putting them on the path to revitalization by communities [4,5]. The GLWQA identifies AOCs as “geographic areas that fail to meet the general or specific objectives of the agreement where such failure has caused or is likely to cause impairment of beneficial use of the area’s ability to support aquatic life.” [6]. When the program was initiated in 1987, 43 AOCs were identified, 31 of which were entirely or partly in U.S. waters. Most AOCs are located near population centers, are within the Great Lakes coastal zone, and are degraded by legacy contaminants including heavy metals and persistent organic pollutants [7]. The program recognizes 14 distinct beneficial use impairments (BUIs; Figure 1). In brief, a beneficial use impairment is any change (chemical, physical, or biological) to an area which causes significant environmental degradation. Nearly all AOCs have multiple impaired beneficial uses, which arise from multiple causes including sediment and water contamination, habitat loss, excess nutrients and sediment inputs, and improperly functioning storm or sewer systems [8]. The AOC program’s goal is to remove BUIs through contaminated sediment remediation, aquatic habitat restoration, or both. To delist an AOC, delisting targets and corresponding management actions such as specific remediation or restoration projects are established by local advisory groups through a remedial action plan (RAP) [9]. The AOC program is among the largest environment clean-up programs in the world, having spent nearly $23 Billion US dollars between 1985 and 2019 [4]. Ecological restoration efforts within AOCs have been associated with increased ecosystem services [10] and social benefits [11], including increased property values, recreation, and waterfront development [7,12,13], as well as the potential for improved human health and wellbeing [14].

Four BUIs recognize direct connections between ecological health and human health or behaviors: Restrictions on Fish and Wildlife Consumption, Restrictions on Drinking Water Consumption, Taste and Odor Problems, or Beach Closings. However, most of the BUIs are indirectly linked to human health or wellbeing, often via the loss of ecosystem services (i.e., Degradation of Aesthetics, Degraded Fish and Wildlife Populations). For these, evaluating impacts on human health is difficult. There is a need to study BUIs and their connections to human health regardless of the directness of that connection because cumulative and indirect effects can have important impacts on human and ecological health and wellbeing [15,16]. Understanding how and why BUIs impact human health can help guide future restoration efforts, elucidate understandings of physiological and psychological pathologies, and inform how interactions with environments can impact these pathologies.

Despite recognition of the potential for improved human health, to our knowledge, there is no research demonstrating a direct effect between improved environmental quality and ecological health (regardless of the BUI addressed) on human health. Nevertheless, human health is among the most preferred outcomes of remediation and restoration efforts in AOCs [17]. The difficulty of linking environmental quality and human health within AOCs is only compounded when looking at BUIs that indirectly effect health, such as the degradation of aesthetics.

### 1.1. What Does It Mean to Be “Aesthetically Degraded”?

When classifying an area as ecologically degraded there are conventional approaches, such as indices of biotic integrity. These approaches are typically ecosystem or habitat-specific, and measure qualities, such as the species or abundance of wildlife or fish, species or density of trees or other vegetation, associated biodiversity within specific taxonomic groups, or soil or water chemistry. To support classification, these approaches are often designed to measure the relative number of species that either prefer high-quality conditions or low-quality conditions [18]. However, there is a distinction between what is ecologically degraded and what people find to be aesthetically degraded. For example, log floats and other deadfall debris in waterways are often considered aesthetically displeasing but can be an ecological benefit to wildlife [19]. The majority of the complaints around aesthetically degraded sites relate to water quality and perceptible pollution, in particular water odor, color, clarity, litter, sheens, and foam [10,20,21]. For green spaces, the focus is often on litter, trail maintenance, and perceived naturalness of the space [22,23,24].

For the AOC program, when the Degradation of Aesthetics BUI was written, it was designed to address slicks or foam from industrial wastewater discharge. Later, as many of these discharges were addressed through wastewater permitting, the BUI did not apply to a contemporary environmental state [25]. Subsequently, the BUI was revised to address a broader suite of conditions, including unnatural odors, objectionable deposits, or unnatural colors or turbidity [26]. However, within AOC-specific RAPs, it also can address litter and debris [27].

### 1.2. What Is the Aim of This Review?

We conducted a comprehensive literature review to summarize the current literature that addresses aesthetics in an environmental exposure with human health outcomes, as well as to determine the utility of future studies. While much of the underlying literature included here is based in the blue and green space research communities where large systematic reviews have already been conducted, our aim in conducting this review is to outline the interest in degradation of aesthetics and human health and lend insight on how studies could be designed to address this understudied topic.

## 2. Materials and Methods

### 2.1. Overview

The process of this systematic review followed procedures similar to those used in large systematic reviews of the blue and green space literature [28,29,30]. We started by identifying key search terms and the databases/search engines most compatible. We next developed a set of screening criteria to evaluate each article for inclusion and exclusion from the review. Articles were then screened, and data extracted for synthesis of this review. More in-depth analysis of each included article is presented in Appendix A and narrative summaries.

### 2.2. Identification of Relevant Search Terms and Engines

To identify key search terms relevant to degradation of aesthetics and human health impacts, we identified keywords from both aesthetics and loss of habitat literature. The aesthetics literature yielded terms related to water quality such as turbidity, water color, and odor, whereas the loss of habitat literature yielded terms related to recreational use and services. We also examined literature that theorized connections between aesthetic degradation or loss of habitat with human health for possible health outcome-related terms. These studies used terms such as aesthetic value, visual quality, or an aesthetic quality index to capture psychological perceptions of natural spaces. The psychological terminology throughout the literature also yielded terms describing health-outcomes, including depression, anxiety, and other mood disorders. We combined these literature-specific efforts to develop an encompassing list of relevant search terms (Figure 2) for both exposures and outcomes related to degradation of aesthetics. We conducted literature searches through 31 July 2021 with the Environmental Protection Agency’s (EPA)s Health & Environmental Research Online (HERO) librarians within PubMed and Web of Science databases to identify relevant articles.

### 2.3. Screening Criteria

To screen articles, we developed three necessary criteria and one optional criterion. The first necessary criterion was that the article’s exposure be aesthetic related. For example, we encountered studies that addressed loss of habitat, urban greenspaces, as well as other studies of the aesthetics of an environment. The second necessary criterion was each article had at least one human health outcome. This could be either a psychological or physiological outcome and had to demonstrate an attempt to directly relate the aesthetics of an environmental exposure to human health. The third criterion was that the article presented an original study. We also examined relevant review studies returned from the search for original research that had not been directly captured. Our optional criterion was that the study have been conducted in a location within the Great Lakes region.

### 2.4. Screening and Data Extraction

Screening of the articles collected by the EPA HERO librarians was conducted using the web-based systematic review software SWIFT-ActiveScreener (SWIFT-AS). A total of 1271 articles were initially extracted by EPA HERO librarians and imported into SWIFT-AS for two-phase screening. Phase one, title and abstract screening, was divided among the authors with each title and abstract screened by two reviewers. Reviewers used the above-mentioned criteria to include or exclude articles. All conflicts in title and abstract screening were reconciled by a consensus of all reviewers before moving to the second phase of screening, full text screening.

After title and abstract screening arbitration, a total of 68 articles were included for full text screening. Three additional studies were added for full text screening from author knowledge. Each of these articles was screened by two reviewers with the same criteria used for title and abstract screening. Again, if any conflicts occurred in screening, all reviewers met to reconcile based on consensus. After arbitration, a total of 19 articles were included for final review and data extraction (Figure 3).

The final 19 articles included for review underwent data extraction in which relevant information was obtained. Basic article information included title, author, journal, and publication year. Information regarding study design, population studied, study years, study country, and area type (i.e., urban, coastal, inland, etc.) were also extracted, as well as statistical methods, sample size, covariates, effect measures, their reported values, and confidence intervals. Each exposure, its description, units, and increments were extracted, as were every outcome, its description, and assessment method. Lastly, any subsets, stratifications, or effect modifiers were noted in extraction along with limitations identified by the article authors.

We evaluated studies across the domains of population selection, exposure, outcome, and analysis each with a number of evaluation questions to determine if any given study adequately addressed the criteria. Evaluation questions are presented in Appendix A. Each study was evaluated against these criteria by at least two of the reviewers and assigned a binary (Y/N) score for having met the criterion or not. Discrepancies were adjudicated after arbitration by a third reviewer. It is important to note that these evaluation questions were not directed towards risk of bias or overall study quality, but to how well the study potentially addressed the question of an association between the degradation of aesthetics, or its removal, and human health.

## 3. Results

Study characteristics are presented in Table 1. All but one of the 19 included studies were cross-sectional in study design, the exception was a crossover intervention [31]. While the earliest population was from 1999 [32], the earliest publication date was 2016, with 10 studies published between 2016–2019 and eight published in 2020–2021, highlighting the relative recentness of research near the topic of interest. Study locations were primarily in North America (*n* = 7; [31,32,33,34,35,36,37]), and China (*n* = 5; [38,39,40,41,42]). Other studies were done in European populations (*n* = 4; [43,44,45,46]), Australia and New Zealand (*n* = 2, [47,48], and one study was performed in Guyana [49]. The size of population studies varied widely, from 23 individuals [31] in the crossover intervention to over 100,000 in claims-based research [48]. Most studies examined some measure of greenspace/bluespace characteristics or proximity, including potential access to these. Other included exposure metrics were beach litter, street level built environment characteristics, and naturalness as measured by birdsong and species diversity. Narrative descriptions of key elements for each of the articles and commentary on how the article touched upon the degradation of aesthetics are provided in Appendix A.

To synthesize the overall literature review from the narrative reviews, we created a study evaluation chart (Table 2) using our domain and criteria questions. The domains of population selection, exposure, outcome, and analysis broadly cover an evaluation of each study included in this review among these areas. It should be noted that none of the included studies were specifically designed to answer a degradation of aesthetics question explicitly. Yet, each of the included studies touched upon overlapping topics and/or methods. Mainly, studies did well in regard to population selection, outcome, and analysis criteria that were not explicitly about aesthetics. The domain most often lacking was exposure, as most studies focused on presence/absence or distance to green and blue space and not more detailed information on space aesthetics and quality. The evaluation domains and questions we utilized are not exhaustive of the possible ways to evaluate these studies in the context of this review but offer an initial framework to begin to evaluate the question of aesthetic degradation in human health research.

## 4. Discussion

We reviewed the available literature on degradation of aesthetics and human health to determine the current state of the science. We found 19 studies that met enough criteria to be included and even approached the topic of interest. This handful of articles demonstrated current attempts to capture elements of aesthetic quality and their importance to generalized human health endpoints. The studies identified for this review continue to support links between green space and various metrics of health, with additional evidence for blue space benefits on health. In the two studies specifically examining beach litter, though people surveyed were more concerned with litter’s impacts on wildlife than health, litter was associated with injuries across demographic groups and especially in children. Perceived naturalness of green space, plant diversity, and green space maintenance were also self-reported to be beneficial to health in surveys. The small but well done crossover waterfront promenade intervention study demonstrated the potential benefits of aesthetic improvements to public spaces, including simple additions of seating and viewing frames with historical images for comparison [31]. Despite these findings, no studies to date adequately address questions surrounding the beneficial use impairment degradation of aesthetics and how improving either environmental quality (remediation) or ecological health (restoration) efforts have impacted the health of those communities. Much has been written tangential to degradation of aesthetics, indicating scientific interest, and based on the publication of included studies the interest in the aesthetic quality of environmental spaces is growing.

Among the 19 included articles, we identified multiple shortcomings relative to our interests. Most of these papers focused on typical exposures to green and blue spaces with relatively non-specific health outcomes (i.e., wellbeing). We also found a dearth of studies conducted in the Great Lakes region, an area heavily invested with binational restoration programs. This was surprising given the long-lived efforts in the Great Lakes to address several beneficial use impairments. The GLWQA and the designation of AOCs lends itself to study of any of the beneficial use impairments, including degradation of aesthetics. Lastly, the study designs and analytic approaches of the included articles were not fully sufficient to accurately assess the impact of aesthetic quality, degradation, and human health outcomes. The missing pieces among the included articles were longitudinal design and robust ecological assessment.

While none of the included articles would definitionally meet criteria for a robust study evaluating degradation of aesthetics and a specific human health outcome, these studies are at least tangentially related to degradation of aesthetics by adding to a growing body of literature on the influence ecosystems goods and services exposure has on human health. Further, green and blue space studies often do have a gradation of exposure, even if it is simply more green or blue space is better than less. While this is not a robust assessment of environmental quality, it does make the implicit assumption that more natural spaces may confer greater health benefit. Several of the included studies, and notably the works of Fisher et al. and Campbell et al. [48,49], took this assumption a step further by trying to quantify naturalness or rank spaces based on perceived quality of the space by users and then link these perceptions to human health outcomes.

It was encouraging to see interest developing in the questions surrounding degradation of aesthetics and human health. Several of the articles reviewed indicate an implicit assumption that healthier and more aesthetically pleasing environments may improve human health. For example, the beach litter studies work under the assumption that less littered beaches are both more aesthetically pleasing, healthier, and safer for beach users. This is likely a safe assumption but can be more nuanced when talking about non-anthropogenic beach litter, such as seaweed and driftwood, that accumulate naturally on beaches but are then often cleared away before beaches open for public use. Still, studies that are looking at perceptions around natural spaces are contributing in a meaningful way by capturing, at least tangentially, people’s aesthetic preferences. These perceptions and values can be compared to other value systems and if measures of ecological health are collected, other ecosystems. Studies measuring both aesthetics or perceptions and ecosystem health may be able to inform restoration policy and future restoration efforts or amend current efforts. Yet, the balance between what people find aesthetically pleasing and what may be ecologically important for the wildlife is a piece often not evaluated in studies. Additionally, it is important to study areas that have been degraded by industrialization, such as the Great Lakes region, and have been successfully restored to see how human health has been impacted. This opens a new area of research to be explored going forward in line with the OneHealth approach.

There are several issues impeding most blue or green space studies from evaluating the relationship between aesthetic degradation and human health. The first issue is the assumption that a “natural” space (however defined) is equivalent to the quality of the space, and the defining of what constitutes high quality space. The second issue is a lack of controls. Most green or blue space studies only look at individuals who use the park/space and do not account for those individuals that do not go to the park/space. These two issues preclude any robust quantitative assessment of aesthetic degradation and human health but do not stop much speculation regarding a connection. However, well-designed studies are attainable.

Cross-sectional designs are still important to improving our understanding of human-environment interactions. However, to more accurately assess changes to the environment, longitudinal studies will need to be conducted. Ideally, a longitudinal study could assess a space or spaces at an early timepoint, measuring specific human health outcomes in a well-defined population, and then reassess the same space or spaces at a later timepoint to account for either degradation or remediation. Assessing degradation and remediation requires the second missing piece, robust ecological assessment. While Fisher et al. [49] did assess bird song and species diversity as an ecological health proxy, few of the included articles went to such lengths. To accurately assess degradation and remediation, future studies will need to conduct robust ecological assessments, capturing features, such as water quality, soil quality, wildlife health, and biodiversity. Lastly, many studies evaluated human health using generalized self-report measures (i.e., wellbeing). This method fits well in cross-sectional design and provides important insight, but makes hypothesis generation of specific mechanism of action for human–environment interaction difficult. Wu and Jackson [36] did use the specific outcome of Alzheimer’s Disease diagnosis, and Roe [31] also used a specific measure with heart rate variability. By using more specific human health outcomes and robust ecological assessment, we may begin to gain a deeper understanding of how changes to natural spaces impact human health and perceptions of aesthetics.

We interpret that the lack of studies specifically addressing degradation of aesthetics is not due to a lack of interest on the topic, but the difficulty in studying it. A key hurdle to conducting aesthetics, as well as green or blue space, research is defining the space and the terms used to describe it. The problem has been noted extensively in the green space research community [50]. Much of the literature that mentions aesthetic degradation or related topics (i.e., biodiversity, naturalness, etc.) occurs within the blue and green space research communities [23,24,51,52,53,54,55,56,57,58,59,60,61,62,63,64,65]. This is not surprising given the concerns around use and factors that influence space use. Some of the noted factors that appear to influence use are accessibility, safety, and perceptions about the naturalness of the space [59,62,65,66]. It is often through this lens of perceptions about green or blue spaces that peoples’ understanding of aesthetic degradation is most revealed. Much of this perception literature touches on the topic of aesthetic degradation but fails to draw a direct link to any human health outcome [53,58,59,66]. Without more detailed exposure assessment, it is difficult to parse out who is utilizing the space and how they utilize it. This difficulty in defining exposure also impairs researchers’ abilities to make conclusions about the immediacy and magnitude of effects from green or blue space exposure. The spaces could confer a diffuse passive benefit to those residing near them, as seems indicated in numerous air pollution mitigation studies [67,68,69,70], or possibly have a direct benefit to those who utilize the space for physical activity and mental health [71,72,73,74]. Of specific concern to addressing ecological degradation is the phenomena that people avoid negative exposures. Park goers will avoid a space that they perceive as either aesthetically displeasing or possibly harmful to them and we cannot begin with the assumption that the landscape or space is neutral with respect to someone’s choice. Additionally, remediation efforts may not have perceptible impacts if, for example, the work is focused in sediment that is at the bottom of a river, lake, or estuary and out of view. Moreover, over years or decades, population shifts such as gentrification may give the appearance of a newly restored green or blue space conferring health benefits, when these apparent benefits are derived from the new, often more wealthy residents.

Future studies designed to investigate the relationship between degradation of aesthetics and human health will need to build on three key components: longitudinal (or equivalent) studies with multiple data points over time, data to describe quality of blue or green or other ecological space, and measuring health outcomes in such a way that can be related to specific pathologies. First, longitudinal and comprehensive retrospective study designs will need to be deployed to measure the changes made to areas identified as aesthetically degraded. These changes could include changes to space quality (i.e., degraded to non-degraded or vice versa), changes to space area/orientation (i.e., expansion or restructuring), or changes to space surrounding population composition (i.e., gentrification). This will require data collection and curation on a scope and scale not typical of smaller green and blue space studies. Furthermore, ecological health data, such as standardized indices of biotic integrity or vegetation surveys, will need to be collected to establish distinct conditions for when an area moves from degraded to non-degraded (or vice versa) from an ecological perspective. These indices, in combination with qualitative assessments of perceptions, may help to remove some of the ambiguity around aesthetics and provide a foundation for more systematic assessments of degradation of aesthetics. Lastly, emphasis needs to be placed on measuring discrete human health outcomes to better capture potential modes of action for the impacts of aesthetics on human health. Preferably, this could be done using more robust methodologies, such as clinical and biomarker data. Self-report and survey-based information may still be necessary, but every effort to ensure the validity, reproducibility, and transportability of these methods should be made in future studies. Further, the Great Lakes Areas of Concern are well suited for studying degradation and remediation, and emphasis should be placed on conducting analyses in surrounding communities. To accomplish these kinds of studies, interdisciplinary collaboration with clinicians, epidemiologists, ecologists, social scientists, key stakeholders, and community members will be required.

## 5. Conclusions

In this review, which sought to assess the available research connecting the degradation of aesthetics BUI to human health, we found a body of literature that is sparsely populated and of limited ability to address the specific questions of interest, though supporting linkages between more general metrics of environmental spaces and health. Many blue and green space studies have attempted to capture elements of aesthetic quality or degradation (i.e., perceptions about parks and spaces). We found relatively few studies that connected these with health outcomes. Of those that did, there was evidence that poor aesthetics can have a detrimental health effect and that more natural environments can have a mitigating or protective health effect. Unfortunately, these studies were not equipped or designed to capture longitudinal shifts in ecological health or populations and so could not accurately assess degradation or improvement of aesthetics. Though we were unable to do so, future reviews of this topic will hopefully be able to narrow their focus to studies that directly address the quality of aesthetic space.

It is apparent from this review that the wider literature has demonstrated a clear interest in linking human health outcomes to aspects of environmental aesthetics, including using natural areas to improve health. This interest is applicable not only for the GLWQA AOC program, but globally for a variety of potential aspects of aesthetic quality of environmental spaces. In order for policy and decision makers to make the most appropriate decisions for their efforts toward improving the quality of spaces and recreational areas, additional research directly addressing these questions is needed. This review, in addition to attempting to summarize the current available literature related to the degradation of aesthetics, has helped outline that there are opportunities for future research efforts to improve the assessment of aesthetics from either ecological or human perception perspectives, to establish direct and indirect connections between aesthetics and human health, and to link longitudinal changes in aesthetics to human health. Such research has promise to deliver improved measures of health outcomes and a better understanding of relevant pathologies in relationship to ecological restoration and the improvement of aesthetics in natural spaces.

## Figures and Tables

**Figure 1 ijerph-19-06090-f001:**
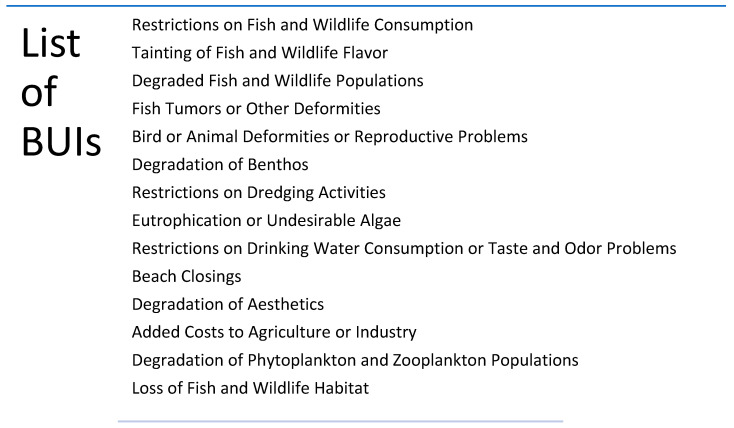
List of all Beneficial Use Impairments.

**Figure 2 ijerph-19-06090-f002:**
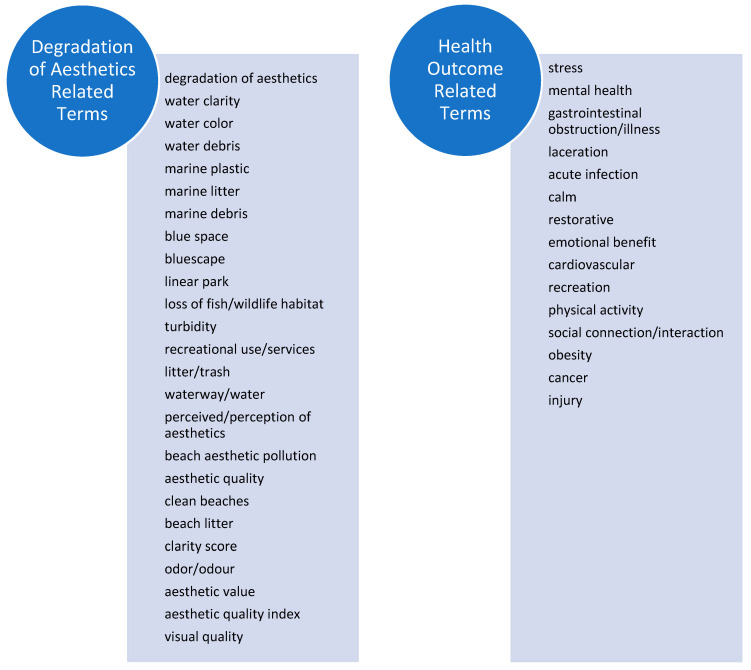
List of search terms, full search strategy and parameters presented in Appendix A.

**Figure 3 ijerph-19-06090-f003:**
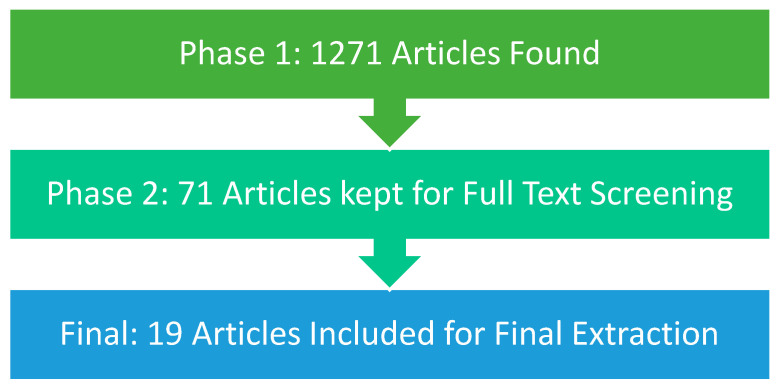
Flowchart of article screening.

**Table 1 ijerph-19-06090-t001:** Summary of data extracted.

Study	Location	Population	Exposure	Outcome & Sample Size	Covariates
The Effects of Naturalness, Gender, and Age on how Urban Green Space is Perceived and Used Sang, 2016, Urban Forestry & Urban Greening [45]	Gothenburg, Sweden	Households living close to six different urban green spaces in 2016	Perceived naturalness based on six areas of diverse character (urban park, woodland, nature area, residential, allotment) assessed by survey	Self-report wellbeing assessed by WHO (ten) well-being index *n* = 1347	age gender
Residential Green Space and Birth Outcomes in a Coastal Setting Glazer, 2018, Environmental Research [33]	Rhode Island, United States	Births occurring at Women & Infants Hospital of Rhode Island, to >17 years at delivery, singleton, living within RI, GA 22–44, birthweight 500–5000 g, with data on covariates from 2002–2004 & 2006–2012	Residential distance to and buffer density of green and blue spaces assessed by NDVI and linear distance	Preterm, birthweight, and small for gestational age assessed by birth record and standard cut points (<37 weeks, grams, birth weight < 10th percentile) *n* = 61,460	maternal age, race, number of prenatal visits, maternal education, marital status, insurance coverage, tobacco use, neighborhood SES, gestational age at birth, town of residence, distance to major roadways
The Association Between Natural Environments and Depressive Symptoms in Adolescents Living in the United States Bezold, 2018, Journal of Adolescent Health [32]	United States	GUTS (Growing Up Today) adolescents cohort 1999	Residential proximity and buffer density of green and blue space assessed by NDVI and linear distance	Depressive symptoms assessed by McKnight risk factor survey *n* = 9385	race, grade level, age, gender, household income, father’s education, maternal history of depression, median tract income, home value, percent tract white, tract college education, region of country, urban/rural, PM_2.5_ average for July 1999
Natural Environments and Suicide Mortality in the Netherlands: a Cross-sectional, Ecological Study Helbich, 2018, The Lancet Planetary Health [44]	Netherlands	National suicide register from 2005–2014	Proportion of greenspace/bluespace and coastal proximity per municipality assessed by Dutch land-use database	Registered suicide deaths assessed by death certificate *n* = 16,105	gender, divorce, unemployment, housing values, distance to nearest GP, voter alignment, urbancity
Are our Beaches Safe? Quantifying the Human Health Impact of Anthropogenic Beach Litter on People in New Zealand Campbell, 2019, Science of the Total Environment [48]	New Zealand	ACC insurance claims from 2007–2016	Reported insurance claims related to injury from beach litter per region	Injury type noted in insurance claim *n* = 161,261	age, gender, ethnicity, location
Effects of Freshwater Blue Spaces may be Beneficial for Mental Health: A First, Ecological Study in the North American Great Lakes Region Pearson, 2019, PLoS ONE [34]	Michigan, United States	Michigan residents in the MIDB during 2014	Proximity/coverage of bluespace assessed by linear distance and zip code overlap	MIDB reported anxiety/mood disorder *n* = 30,421	age, gender, median income, population density
Human Health Impacts from Litter on Beaches and Associated Perceptions: A Case Study of ‘clean’ Tasmanian Beaches Campbell, 2016, Ocean & Coastal Management [47]	Tasmania, Australia	Tasmania beach users from 2010–2011	Frequency of attendance to any of nine beaches across Tasmania assessed by survey	Survey self-reported injury occuring at beaches related to litter *n* = 173	NA
Using Deep Learning to Examine Street View Green and Blue Spaces and their Associations with Geriatric Depression in Beijing, China Helbich, 2019, Environment International [39]	Beijing, China	Elderly population residing in Haidian district during 2011	Neighborhood green/blue space measured by Landsat, NDVI,NDWI, and street view neighborhood green/blue space measured by Landsat, NDVI,NDWI, and street view	Depressive symptoms assessed by geriatric depression scale (GDS-15) *n* = 1190	gender, age, education, marital status, ADL score, multiple chronic diseases, air pollution
Designing Urban Green Spaces for Older Adults in Asian Cities Tan, 2019 [38]	Hong Kong and Tainan	Elderly population of Hong Kong and Tainan 2016–2018	Attendance to one of 31 small scale urban greenspaces	General health survey *n* = 326	NA
Neighbourhood Blue Space, Health and Wellbeing: The Mediating role of Different Types of Physical Activity Pasanen, 2019, International Journal of Environmental Research and Public Health [46]	England, United Kingdom	English households from 2008–2012	Coastal proximity to bluespace and present/absent freshwater bluespace assessed by land use database and linear distance	Self-reported general health assessed by standardized health survey *n* = 21,097	quantity/quality of blue and greenspace, urban/rural, deprivation index, age, gender, education, marital status, household income, employment, car availability, number of children, long-term illness, year
The neighborhood effect of exposure to blue space on elderly individual’s mental health: A case study in Guangzhou, China Chen & Yuan, 2020, Health and Place [40]	Guangzhou, China	Elderly adults sampled from 18 neighborhoods in 2018	Remote sensed neighborhood blue space (characteristics, nearness, visitation)	Self-reported mental health assessed by 36-item Short Form Health Survey *n* = 966	age, gender, education, marital status, hukou status, monthly household income, employment information
Green and Blue Space Availability and Self-Rated health among Seniors in China: Evidence from a National Survey Lin & Wu, 2021, International journal of environmental research and public health [41]	China	Chinese Social Survey respondents aged 60 years or more from 2011	Neighborhood green and blue space assessed by linear distance and buffer area coverage via NDVI/Lansat, Inland Surface Water Dataset	Self-reported overall health assessed via Chinese Social Survey *n* = 1773	age, marital status, ethnicity, insurance, lifestyle education, household registration location, occupation, income, assets, distance to major roadway, population density, GDP production per km^2^
The effect of urban nature exposure on mental health—a case study of Guangzhou Liu, 2021, Journal of Cleaner Production [42]	Guangzhou, China	Survey respondents from 23 residential communities across Guangzhou from 2020	Nearest park and network distance to park and buffer area coverage of blue space using Open Street Map	Self-reported mental health assessed by the Mental Health Inventory *n* = 933	age, gender, education, income, education, income, occupation, marital status, and residence location, urban, life events
General health and residential proximity to the coast in Belgium: Results from a cross-sectional health survey Hooyberg, 2020, Environmental research [43]	Belgium	Respondents of the Belgian Health Interview Survey as of 2013	Network distance to the coast assessed via Open Street Map	Self-reported general health via Belgian Health Interview Survey *n* = 60,939	age, sex, chronic disease, body mass index, employment, income, smoking, urbanization, year, season, green space, blue space
Different types of urban natural environments influence various dimensions of self-reported health Jarvis, 2020, Environmental research [37]	Vancouver, Canada	Respondents of the Canadian Community Health Surveys from 2013–2014	Buffer landcover type via 2008–2015 LiDAR and aerial photography plus access to public greenspace via presence of greenspace within 300 m	Self-reported general health and mental health assessed via the Canadian Community Health Survey *n* = 2,183,170	age, gender, race/cultural background, education, household income, urbancity
Cross-sectional association between the neighborhood built environment and physical activity in a rural setting: the Bogalusa Heart Study Gustat, 2020, BMC public health [35]	Bogalusa, United States	Questionnaire respondents of the Bogalusa Heart Study from 2012–2013	Built environment scores for buffer area surrounding residence assessed via the Rural Active Living Assessment and Google Street View	Physical Activity Questionnaire data weekly metabolic equivalent minuets for leisure, transport, and total physical data. *n* = 1245	age, race, body mass index, education, income, smoking, alcohol consumption, percent census block below poverty, population density
Perceived biodiversity, sound, naturalness, and safety enhance the strotive quality and wellbeing benefits of green and blue space in a neotropical city Fisher, 2021, Science of the Total Environment [49]	Georgetown, Guyana	Survey respondents from 15 natural sites across Georgetown in 2019	Live birdsong and species diversity assessed via recordings and photography	Self-reported wellbeing assessed via the Positive and Negative Affect Schedule *n* = 409	age, ethnicity, religion, education, household income, location of residence
Greenspace Inversely Associated with Risk of Alzheimer’s Disease in the Mid-Atlantic United States Wu & Jackson, 2021, Earth [36]	United States	Centers for Medicaid and Medicare recipients 65 years and older residing in Mid-Atlantic Region from 1999–2013	Landcover type assessed via aerial photography and classified at the zipcode level	Diagnosis of Alzheimer’s Disease via ICD-9 code in patient record. *n* = 109,405	monthly average PM_2.5_, percent greenspace, percent water area, houshold income, zip code area, population density, road density
The Restorative Health Benefits of a Tactical Urban Intervention: An Urban Waterfront Study Roe, 2019, Frontiers in Built Environment [31]	West Palm Beach, United States	Pedestrians along West Palm Beach Promenade Spring 2017	Crossover trial comparing normal promenade conditions (i.e., no changes) to one with minor aesthetic changes	Real-time heart rate variability, subjective mood, and perceived restorativeness assessed via wearable device and surveys *n* = 23	NA

Abbreviations (in order of appearance): WHO, World Health Organization; RI, Rhode Island; GA, Gestational Age; g, Grams; NDVI, Normalized Difference Vegetative Index; SES, Socioeconomic Status; PM_2.5_, Particular Matter (≤2.5 μm in diameter); GP, General Practitioner; ACC, Accident Compensation Corporation; MIDB, Michigan Inpatient Database; NDWI, Normalized Difference Water Index; ADL, Activities of Daily Life; GDP, Gross Domestic Product.

**Table 2 ijerph-19-06090-t002:** Study Evaluation Chart.

Domain	Criteria	Sang 2016	Glazer 2018	Bezold 2018	Helbich 2018	Campbell 2019	Pearson 2019	Campbell 2016	Helbich 2019	Tan 2019	Pasanen 2019	Chen 2020	Lin & Wu 2021	Liu 2021	Hooyberg 2020	Jarvis 2020	Gustat 2020	Fisher 2021	Wu & Jackson 2021	Roe 2019
**Population selection**	Is the population studied well suited for studying exposure to aesthetic degradation?	Y	Y	Y	N	Y	Y	Y	Y	Y	Y	Y	Y	Y	Y	Y	Y	Y	Y	Y
	Is population selection, recruitment, inclusion/exclusion, etc., given in sufficient detail?	Y	Y	Y	Y	Y	Y	Y	Y	Y	Y	Y	Y	Y	Y	Y	Y	Y	Y	Y
	Are there sufficient numbers of included population to observe associations?	Y	Y	Y	Y	Y	Y	N	Y	Y	Y	Y	Y	Y	Y	Y	Y	Y	Y	N
**Exposure**	Were there quantitative approaches to describe the aesthetic condition?	Y	N	N	N	N	N	N	N	N	N	Y	N	N	N	N	Y	Y	N	Y
	Was aesthetic condition defined, and captured in a way consistent with that definition?	Y	N	N	N	N	N	N	N	N	N	Y	N	N	N	N	Y	Y	N	Y
	Are sub-types of habitats and associated areas described?	Y	Y	N	N	N	N	N	N	N/A	N	N	Y	N	N	Y	N	Y	N	N
	If the study examined green/blue space, was this examined beyond the presence or absence of that space?	Y	N	N	N	N/A	N	N/A	N	Y	N	Y	N	N	N	N	N	Y	N	Y
	Is the exposure environment/controls appropriate to test the experience? Is there an exposure control/negative exposure?	Y	N	N	N	N	N	N	N	N	N	N	N	N	N	N	Y	N	N	Y
**Outcome**	Is there measurement of a health outcome as opposed to an assessment of risk or hazard?	Y	Y	Y	Y	Y	Y	Y	Y	Y	Y	Y	Y	Y	Y	Y	Y	Y	Y	Y
	Is there a clear mode of action laid out for exposure to impact health?	N	N	Y	Y	Y	Y	Y	Y	Y	Y	Y	Y	Y	Y	Y	Y	Y	Y	Y
	Is the outcome measured appropriately? Is the outcome measure specific and unlikely to be misclassified? Is there a temporal component to the outcome measure in regard to the exposure?	Y	Y	Y	Y	Y	Y	Y	Y	Y	Y	Y	Y	Y	Y	Y	Y	Y	Y	Y
**Analysis**	Are appropriate confounders considered and accounted for?	N	Y	Y	Y	Y	N	N	Y	Y	Y	Y	Y	Y	Y	Y	Y	Y	Y	Y
	Are the methods used in modeling appropriate?	Y	Y	Y	Y	Y	Y	Y	Y	Y	Y	Y	Y	Y	Y	Y	Y	Y	Y	Y
	Does the study design support whether the effect is based on relative state of physical space or absolute quality of space?	N	N	N	N	N	N	N	N	Y	N	Y	N	N	N	N	Y	Y	N	Y

## Data Availability

Data not presented in the article are available upon request from the corresponding author.

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
