# Peer review of "Beneficial Use Impairments, Degradation of Aesthetics, and Human Health: A Review"

_ijerph, 2022, doi:10.3390/ijerph19106090_

Round 1

Reviewer 1 Report

(1) There are many similar review papers in the literature, and the authors should clarify the differences between them in the introduction. In addition, the authors should clearly state the excellence of this study in the introduction.

(2) Authors should clarify the screening process of the 19 papers used. Methodologies should be compared and justified with similar review papers in the literature. Please perform a more in-depth analysis of the 19 papers.

(3) Table 1 occupies an excessively large amount in this study. Table 1 should be rearranged as an appendix, and a more summary of it should be described in the text. Also, discussion of the findings of the study is somewhat lacking. Please discuss Table 2 in more depth and provide additional implications and insights in the discussion part.

(4) In conclusion, please re-summarize the theoretical and practical contribution of this study.

Author Response

Thank you for your time and feedback. We hope we have addressed your comments. 

  • There are many similar review papers in the literature, and the authors should clarify the differences between them in the introduction. In addition, the authors should clearly state the excellence of this study in the introduction.

Response: We have added further clarification for the aim of this review and how it relates to existing reviews of the literature. “We conducted a comprehensive literature review to summarize the current literature that addresses aesthetics in an environmental exposure with human health outcomes, as well as to determine the utility of future studies. While  much of the underlying literature included here is based in the blue and greenspace research communities where large systematic reviews have already been conducted, our aim in conducting this review is to outline the interest in degradation of aesthetics and human health and lend insight on how studies could be designed to address this understudied topic.”

  • Authors should clarify the screening process of the 19 papers used. Methodologies should be compared and justified with similar review papers in the literature. Please perform a more in-depth analysis of the 19 papers.

Response: We have added a section to the methods to better orient the reader to the overall screening process of this review, how it relates to over reviews of the literature, and reiterated the location of narrative summaries and supplemental files for those seeking more in-depth analysis of any specific article included.

Overview

                The process of this systematic review followed procedures similar to those used in large systematic reviews of the blue and green space literature [27-29]. We started by identifying key search terms and the databases/search engines most compatible. We next developed a set of screening criteria to evaluate each article for inclusion and exclusion from the review. Articles were then screened, and data extracted for synthesis of this review. More in-depth analysis of each included article is presented in supplemental documents and narrative summaries.”

(3) Table 1 occupies an excessively large amount in this study. Table 1 should be rearranged as an appendix, and a more summary of it should be described in the text. Also, discussion of the findings of the study is somewhat lacking. Please discuss Table 2 in more depth and provide additional implications and insights in the discussion part.

Response:

While we understand the reviewer’s perspective that Table 1 is large, we feel that this is the most effective way to present individual study details without sacrificing information. This type of presentation is fairly common in systematic reviews and we do not believe it will be an overall formatting issue for a published manuscript. Overall summaries of the information in table 1 are presented in the first paragraph of the results section. Should the editors request it we can move Table 1 to the appendix but would prefer for it to remain as part of the main text.

We have clarified text referring to table 2 in the results as follows, “Mainly, studies did well in regards to population selection, outcome, and analysis criteria that were not explicitly about aesthetics. The domain most often lacking was exposure, as most studies focused on presence/absence or distance to green and blue space and not more detailed information on space aesthetics and quality. The evaluation domains and questions we utilized are not exhaustive of the possible ways to evaluate these studies in the context of this review, but offer an initial framework to begin to evaluate the question of aesthetic degradation in human health research.”

While not explicitly calling back to table 2, the discussion section is largely devoted to the shortcomings and strengths of existing articles and how this relates to conducting studies of degradation of aesthetics and human health in future work.

(4) In conclusion, please re-summarize the theoretical and practical contribution of this study.

Response: We have re-emphasized the contribution of this review in the conclusions. “This review, in addition to attempting to summarize the current available literature related to degredation of aesthetics, has helped outline that there are opportunities for future research efforts to improve assessment of aesthetics from either ecological or human perception perspectives, to establish direct and indirect connections between aesthetics and human health, and to link longitudinal changes in aesthetics to human health. Such research has promise to deliver improved measures of health outcomes and better understanding of relevant pathologies in relationship to ecological restoration and improvement of aesthetics in natural spaces.”

Reviewer 2 Report

nice work, i would add more guidelines on how to improve future studies.

Author Response

We thank you for your time and feedback, and hope we have addressed your comment. 

(1) nice work, i would add more guidelines on how to improve future studies.

Response: Thank you! Small additions for future studies have been added to the discussion “ Lastly, emphasis needs to be placed on measuring discrete human health outcomes to better capture potential modes of action for the impacts of aesthetics on human health; preferably this could be done using more robust methodologies such as clinical and biomarker data. Self-report and survey-based information may still be necessary but every effort to ensure the validity, reproducibility, and transportability of these methods should be made in future studies. Further, the Great Lakes Areas of Concern are well suited for studying degredation and remediation and emphasis should be placed on conducting analyses in surrounding communities. To accomplish these kinds of studies, interdisciplinary collaboration with clinicians, epidemiologists, ecologists, social scientists, key stakeholders, and community members will be required.

Reviewer 3 Report

overall, a needed and valuable contribution to the literature. Much needed attempt at organizing and  assessment in a field of wildly diverse material. 

a surprising lapse: there is no clear definition of BUI, it is assumed that all readers accept the implied connotation!

The study is much needed; material covered is well conceptualized, but-- I don't get it--emphasis is on Great Lakes but material selected belies this!

spell check needed, eg line 196, 

Author Response

Thank you for your time and feedback, we hope we have sufficently addressed your comments.

(1) a surprising lapse: there is no clear definition of BUI, it is assumed that all readers accept the implied connotation!

Response: We thank the reviewer for catching this oversight! A brief definition has been added. “In brief, a beneficial use impairment is any change (chemical, physical, or biological) to an area which causes significant environmental degredation.”

(2) The study is much needed; material covered is well conceptualized, but-- I don't get it--emphasis is on Great Lakes but material selected belies this!

Response: The emphasis of the Great Lakes was tied to the longstanding GLWQA and the vast projects that have improved the area, but which have been understudied in a research context. This review is intended, in part, to motivate researchers to look specifically at the Great Lakes region as a site for studying the effects of environmental degredation and remediation.

To help emphasize this point we have expanded the discussion “We also found a dearth of studies conducted in the Great Lakes region, an area heavily invested with binational restoration programs. This was surprising given the long-lived efforts in the Great Lakes to address several beneficial use impairments. The GLWQA and the designation of AOCs lends itself to study of any of the beneficial use impairments, including degredation of aesthetics… Further, the Great Lakes Areas of Concern are well suited for studying degredation and remediation and emphasis should be placed on conducting analyses in surrounding communities.”

(3) spell check needed, eg line 196, 

Thank you, we have spell checked the manuscript and corrected errors throughout.

Round 2

Reviewer 1 Report

(1) The abstract is too short to guess about the main content and results of the study. Also, please add keywords appropriately.

(2) Authors should show more diverse results, such as showing the names of journals published in 19 articles.

(3) In the conclusion, the authors additionally describe the implications and insights of this study.

Author Response

  • The abstract is too short to guess about the main content and results of the study. Also, please add keywords appropriately.

Abstract has been expanded. We are not sure what is meant by adding keywords appropriately? There is a list of 4 keywords below the abstract as indicated in the instructions/template for IJERPH.

  • Authors should show more diverse results, such as showing the names of journals published in 19 articles.

Journal names have been added to Table 1

  • In the conclusion, the authors should additionally describe the implications and insights of this study.

We have added language to the conclusion to additionally describe implications and insights of this review.